# Two-Dimensional Monte Carlo Filter for a Non-Gaussian Environment

**Xingzi Qiang** [1] , **Rui Xue** [1,*] and **Yanbo Zhu** [2]

1   School of Electrical and Information Engineering, Beihang University, Beijing 100191, China; qiangxingzi@buaa.edu.cn
2   Aviation Data Communication Corporation, Beijing 100191, China; zhuyanbo@buaa.edu.cn
*   Correspondence: xuerui@buaa.edu.cn

**Abstract:** In a non-Gaussian environment, the accuracy of a Kalman filter might be reduced. In this paper, a two- dimensional Monte Carlo Filter is proposed to overcome the challenge of the non-Gaussian environment for filtering. The two-dimensional Monte Carlo (TMC) method is first proposed to improve the efficacy of the sampling. Then, the TMC filter (TMCF) algorithm is proposed to solve the non-Gaussian filter problem based on the TMC. In the TMCF, particles are deployed in the confidence interval uniformly in terms of the sampling interval, and their weights are calculated based on Bayesian inference. Then, the posterior distribution is described more accurately with less particles and their weights. Different from the PF, the TMCF completes the transfer of the distribution using a series of calculations of weights and uses particles to occupy the state space in the confidence interval. Numerical simulations demonstrated that, the accuracy of the TMCF approximates the Kalman filter (KF) (the error is about $10^{-6}$) in a two-dimensional linear/ Gaussian environment. In a two-dimensional linear/non-Gaussian system, the accuracy of the TMCF is improved by 0.01, and the computation time reduced to 0.067 s from 0.20 s, compared with the particle filter.

**Keywords:** nonlinear filter; non-gaussian environment; particle filter; sequence monte carlo





## 1. Introduction

Bayesian inference is one of the most popular theories in data fusion [1–5]. For a linear Gaussian dynamic system, the Bayesian filter can be achieved in terms of the well-known updating equations of the Kalman Filter (KF) perfectly [6]. However, the analytical solution of the Bayesian filter is impossible to be obtained in a non-Gaussian scenario [7]. This problem has attracted considerable attention for a few decades because of the wide application in signal processing [8,9], automatic control systems [10,11], biological information engineering [12], economic data analysis [13], and other subjects [14]. Approximation is one of the most effective approaches for solving the nonlinear/non-Gaussian filter problem.

The linearization of the state model is an important strategy for solving the nonlinear/non-Gaussian filtering problem. The extended Kalman filter (EKF) was introduced to approximate the nonlinear model using the first-order term of the Taylor expansion of the state and observation equations in [15]. In [16], the unscented Kalman filter (UKF) was proposed to reduce the truncation error by introducing the unscented transformation (UT) [17]. The cubature Kalman filter based on the third-degree spherical–radial cubature rule was proposed in [18]. The third-degree cubature rule is a special form of the UT and has better numerical stability in the application of filtering [19]. The Gauss–Hermite filter and central difference filter (CDF) were proposed by Kazufumi Ito and Kaiqi Xiong in [20] and made the Gaussian assumption for the noise model.

Sequential Monte Carlo (SMC) provides another important strategy for the nonlinear/non-Gaussian filtering problem and can approximate any probability density function (PDF) conveniently using weighted particles. Particle Filter (PF) [21,22] is an algorithm derived from the recursive Bayesian filter based on the SMC approach that is used to solve

data/information fusion in a nonlinear/non-Gaussian environment [23]. The SMC approach was introduced in filtering to tackle a nonlinear dynamic system that is analytically intractable. The core idea of PF is to describe the transformation of the state distribution through the propagation of particles in a nonlinear dynamic system, and represent the posterior probability using weighted particles. As a flexible approach for avoiding solving complex integral problems, PF is widely used in the data/information fusion of nonlinear systems, such as fault detection [24], cooperative navigation and localization, visual tracking, and melody extraction. In [25] and [26], EKF and UKF were introduced, respectively, to optimize the proposal distribution for the PF framework. The feedback PF was designed based on an ensemble of controlled stochastic systems [27]. Additionally, because of the advantage of the resampling technique in solving the degeneracy problem, various resampling schemes were proposed in [28–30].

Both strategies are based on Bayesian theory, and approximation is also their main approach for solving the nonlinear filtering problem [31]. However, the perspectives of these two strategies are different, which results in different characteristics. For the first strategy, the Kalman filter (KF) is considered as the representative of Bayesian inference. Obtaining an analytical solution that is close to the real posterior distribution is the trick that the first strategy attempts to solve [32]. Researchers have attempted to approximate complex nonlinear non-Gaussian problems to linear Gaussian problems that can be directly solved using the KF [33]. This approximation inevitably leads to a truncation error. Therefore, many improved algorithms based on the first strategy have been proposed, mainly to reduce the effect of the truncation error on nonlinear filtering [34]. However, it is difficult to accurately obtain the posterior distribution of a nonlinear system [35]. The first strategy is always accompanied by linearization errors. Reducing the influence of the Gaussian assumption of non-Gaussian noise on filtering performance is also a major problem to be considered in the first strategy [36]. For the second strategy, the SMC method is used to solve the difficult problem of integration in Bayesian filtering [37]. Theoretically, this strategy (PF and its improved algorithms) is not constrained by the model of nonlinearity and non-Gaussian environment [38]. However, PF has been plagued by sample degeneracy and impoverishment since it was proposed. Many scholars have proposed several improvement methods to mitigate the two problems [39–41]. Increasing the particle number is an original approach to solve the problems, but it is not very effective because the particle number needs to increase exponentially to alleviate the two problems, which inevitably affects the efficiency of the filter [42]. The two main approaches for solving the two problems are improving the proposal distribution and resampling [43,44]. The improvement of the proposal distribution might greatly alleviate the impoverishment problem to improve the performance of the filter [45]. Hence, related improved algorithms have been widely used in engineering practice, such as EPF and UPF [46]. Resampling, as an important means to alleviate the sample degeneracy of PF has been widely studied by many scholars [47,48]. However, the model used to improve the proposal distribution must be based on some known noise model (such as the Gaussian model in EPF and Gaussian mixture model in UPF), which cannot fully solve the impoverishment problem [49]. The resampling step might alleviate the sample degeneracy, accompanied by the introduction of the resampling error [31]. The main problem of the second strategy is that the particle utilization efficiency is not high, which affects the effect and efficiency of the filtering [40].

Among the existing algorithms, the PF is the most flexible. The main procedure of PF can be roughly summed up as: (1) obtain particles according to a proposed distribution; (2) calculate the prior weights and the likelihood weights according to the process noise model and measurement noise model respectively and mix them; and (3) divide the mixed weights by the corresponding density of the proposed distribution and normalization. After that, the posterior distribution is reflected from these weighted particles. From this procedure, we can observe that the process of select particles is random, but the weights are calculated precisely according to the noise model. This phenomenon might cause random disturbances which could affect the filtering accuracy.

To overcome the aforementioned problem, the two-dimensional Monte Carlo (TMC) method is proposed to improve the efficiency of the sampled particles. Then, the TMC filter (TMCF) algorithm is proposed to solve the non-Gaussian filter problem based on the principle of the TMC method. The main contributions arising from this study are as follows:

(1) The TMC method, as a deterministic sampling method, is proposed to improve the efficacy of particles. Particles are sampled in the confidence interval uniformly according to the sampling interval. Then, the posterior weight of each particle is calculated based on Bayesian inference. Subsequently, any probability distribution can be described by a small number of weighted particles.

(2) A discrete solution to the problem of how to describe a known probability distribution transmitted in a linear or nonlinear state model is proposed. First, a small number of original weighted particles are obtained according to TMC method. Then, the confidence interval of the next time step for a fixed confidence is calculated according to the state model. Some new particles are then set in this confidence interval uniformly in terms of the sampling interval. After that, the weights of these new particles are obtained using a series of calculations based on Bayesian inference. Then, the transferred probability distribution is described by these new weighted particles.

(3) The TMCF algorithm is proposed based on the above two points. The proposed algorithm can be divided into four parts: initialization, particle deployment, weight mixing, and state estimation. The TMC method is used in the initialization step to generate the efficacy weighted particles. Particle deployment solves the problem of state space transfer for a certain degree of confidence and deploys particles in the confidence interval. The weight mixing step achieves the fusion of several arbitrary continuous probability densities in a discrete domain. Some invalid weighted particles are omitted in the particle choice step and the state is estimated using the remaining weighted particles.

(4) The performance of TMCF was verified using a numerical simulation. The results demonstrated that the proposed algorithm with the approach of fewer particles and less computation estimated accuracy better than the PF in linear and Gaussian systems and performed better than the KF and PF in linear and Gaussian mixture noise model.

The outline of this paper is as follows. In Section 2, the problem statement and Bayesian filter are presented. The TMC method is introduced in Section 3. In Section 4, the TMCF algorithm is introduced in detail. The numerical simulation is described in Section 5, and the validity of the proposed framework is demonstrated. In Section 6, the conclusion of this study is presented.

## 2. Problem Statement and Bayesian Filter

### 2.1. Problem Statement

For filtering algorithms introduced in this paper, the state space model is defined as: [37]

$$x_t = f(x_{t-1}) + u_{t-1} \tag{1}$$

$$z_t = h(x_t) + v_t \tag{2}$$

where $x_t \in \Re^{n_x}$ and $y_t \in \Re^{n_y}$ denote the state variable and observation at time step $t$, respectively; $n_x$ and $n_y$ denote the dimensions of the state vectors and observation, respectively; $u_t \in \Re^{n_x}$ and $v_t \in \Re^{n_y}$ denote the system noise and observation noise, respectively; and the mappings $f : \Re^{n_x} \times \Re^{n_u} \mapsto \Re^{n_x}$ and $h : (\Re^{n_x} \times \Re^{n_u}) \times \Re^{n_v} \mapsto \Re^{n_y}$ describe the state transition equation and observation equation, respectively; $z_t$ denotes the observation at time step $t$.

In this paper, $u_t$ and $v_t$ are independent of each other, and the probability distributions of $u_t$ and $v_t$ are $p_u(x)$ and $p_v(x)$, respectively. Meanwhile, the probability distribution of the initial state is known. The goal is to obtain the approximate Bayesian estimation in the filtering process in a nonlinear and non-Gaussian environment.

## 2.2. Bayesian Estimation

Recursive Bayesian filtering provides an effective guide for the real-time fusion of the state equation and observation. The procedure of the Bayesian filter framework can be divided into prediction and update steps as follows:

$$p(\boldsymbol{x}_t|\boldsymbol{z}_{1:t-1}) = \int p(\boldsymbol{x}_t|\boldsymbol{x}_{t-1})p(\boldsymbol{x}_{t-1}|\boldsymbol{z}_{1:t-1})d\boldsymbol{x}_{t-1}$$

$$p(\boldsymbol{x}_t|\boldsymbol{z}_{1:t}) = \frac{p(\boldsymbol{z}_t|\boldsymbol{x}_t)p(\boldsymbol{x}_t|\boldsymbol{z}_{1:t-1})}{p(\boldsymbol{z}_t|\boldsymbol{z}_{1:t-1})}$$

where $p(\boldsymbol{x}_t|\boldsymbol{x}_{t-1})$ denotes the state transition PDF, $p(\boldsymbol{x}_{t-1}|\boldsymbol{z}_{1:t-1})$ denotes the posterior PDF at time step $t-1$, $p(\boldsymbol{x}_t|\boldsymbol{z}_{1:t-1})$ denotes the prior PDF at time step $t$, $p(\boldsymbol{x}_t|\boldsymbol{z}_{1:t})$ denotes the likelihood PDF and

$$p(\boldsymbol{z}_t|\boldsymbol{z}_{1:t-1}) = \int p(\boldsymbol{z}_t|\boldsymbol{x}_t)p(\boldsymbol{x}_t|\boldsymbol{z}_{1:t-1})d\boldsymbol{x}_t \tag{3}$$

For a linear and Gaussian environment, this procedure can be accurately operated by the celebrated KF as the integral problem of Equation (1), and the likelihood probability $p(\boldsymbol{z}_k|\boldsymbol{x}_k)$ can be solved conveniently. For a nonlinear and non-Gaussian environment, it is impossible to solve Equation (1) directly.

## 3. Two-Dimensional Monte Carlo Method

The Monte Carlo approach provides a convenient track inference of the posterior PDF in a non-Gaussian environment. PF is a branch of the family of filter algorithms and is based on the Monte Carlo approach. It is used to process the nonlinear and non-Gaussian system filter problem. Several improved particle filter algorithms exist. The core of the PF approach is to sample particles according to the difference in the proposal distribution. Particles are used to describe the transition of the PDF in the system model. The integration of the observation depends on the likelihood weight. The concept of weight provides the possibility for the application of Monte Carlo to the filtering problem, which plays an important role. In the following, the TMC method is introduced to make full use of the weight and the noise model to enhance particle efficiency.

Suppose $p(x, y)$ is the PDF of a two-dimensional noise model. Its marginal PDF can be expressed as:

$$p(x) = \int_{-\infty}^{+\infty} p(x, y)dy \tag{4}$$

$$p(y) = \int_{-\infty}^{+\infty} p(x, y)dx \tag{5}$$

where $p(x)$ and $p(y)$ denote the marginal PDF of $x$ and $y$, respectively.

The confidence interval $c$ for confidence $1 - \alpha$ can be defined as:

$$c_x = \left[u_{x_1}, u_{x_2}\right] \tag{6}$$

$$c_y = \left[u_{y_1}, u_{y_2}\right] \tag{7}$$

$$c = \left[ \begin{array}{c} c_x \\ c_y \end{array} \right] \tag{8}$$

where:

$$\int_{-\infty}^{u_{x_1}} p(x)dx = \frac{\alpha}{2} \tag{9}$$

$$\int_{u_{x_2}}^{+\infty} p(x)dx = \frac{\alpha}{2} \tag{10}$$

$$\int_{-\infty}^{u_{y_1}} p(y)dy = \frac{\alpha}{2} \tag{11}$$

$$\int_{u_{y_2}}^{+\infty} p(y)dy = \frac{\alpha}{2} \tag{12}$$

Particles $X$ can be set according to sampling interval $dT$ for $c$, as shown in Figure 1. Additionally,

$$dT \equiv \left[dt_x, dt_y\right]^T \tag{13}$$

$$X \equiv \left\{ \begin{bmatrix} x_1 \\ y_1 \end{bmatrix} \quad \begin{bmatrix} x_2 \\ y_3 \end{bmatrix} \quad \cdots \quad \begin{bmatrix} x_n \\ y_n \end{bmatrix} \right\} \tag{14}$$

where $n$ denotes the particle number.

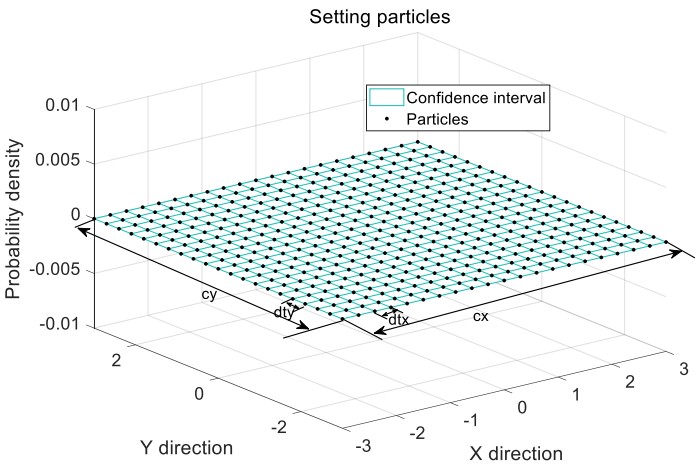

**Figure 1.** Sketch map of setting particles.

The weight of these particles $w$ is calculated as:

$$w \equiv [w_1, \ w_2, \cdots w_n]^T \tag{15}$$

where:

$$w_i = \frac{p(x_i, y_i)}{\sum\limits_{i=1}^{n} p(x_i, y_i)} \tag{16}$$

Then, $\{X, w\}$ is used to describe $p(x, y)$ with the accuracy of $dT$ in the confidence interval $c$ for confidence $1 - \alpha$ discretely. The sketch map of the particles and their weights is shown in Figure 2.

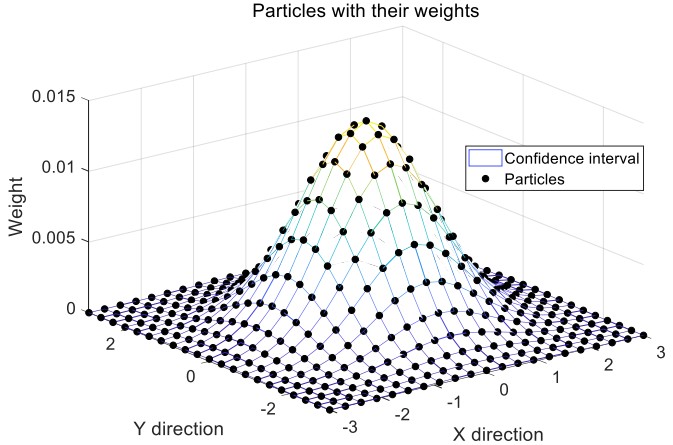

**Figure 2.** Sketch map of particles and their weights.

**Theorem 1.** *When $\alpha \to 0$ and $dT \to 0$, then $n \to \infty$ and*

$$\lim_{n \to \infty} \sum_{i=1}^{n} x_i w_i = \int_{-\infty}^{+\infty} \int_{-\infty}^{+\infty} x p(x, y) dx dy \tag{17}$$

$$\lim_{n \to \infty} \sum_{i=1}^{n} y_i w_i = \int_{-\infty}^{+\infty} \int_{-\infty}^{+\infty} y p(x, y) dx dy \tag{18}$$

**Proof.** Suppose the probability space of $p(x, y)$ is divided into $n$ small squares in terms of $dT$, where $dT \equiv \left[ dt_x, dt_y \right]^T$. When $dT \to 0$, $n \to \infty$ and

$$\lim_{n \to +\infty} \sum_{i=1}^{n} p(x_i, y_i) dt_x dt_y = \int_{-\infty}^{+\infty} \int_{-\infty}^{+\infty} p(x, y) dx dy = 1$$

□

As $dt_x$ and $dt_y$ are independent of $i$,

$$\sum_{i=1}^{n} p(x_i, y_i) = \frac{1}{dt_x dt_y} \tag{19}$$

Hence,

$$\int_{-\infty}^{+\infty} \int_{-\infty}^{+\infty} x p(x, y) dx dy = \lim_{n \to +\infty} \sum_{i=1}^{n} x_i p(x_i, y_i) dt_x dt_y$$

$$= \lim_{n \to +\infty} \left[ \left( \sum_{i=1}^{n} x_i p(x_i, y_i) \right) \frac{1}{\sum_{j=1}^{n} p(x_j, y_j)} \right]$$

$$= \lim_{n \to +\infty} \left( \sum_{i=1}^{n} x_i \frac{p(x_i, y_i)}{\sum_{j=1}^{n} p(x_j, y_j)} \right)$$

$$= \lim_{n \to +\infty} \sum_{i=1}^{n} x_i w_i$$

Thus, $\lim_{n \to \infty} \sum_{i=1}^{n} x_i w_i = \int_{-\infty}^{+\infty} \int_{-\infty}^{+\infty} x p(x, y) dx dy$.

Similarly,

$\lim_{n \to \infty} \sum_{i=1}^{n} y_i w_i = \int_{-\infty}^{+\infty} \int_{-\infty}^{+\infty} y p(x, y) dx dy$.

A simple sample is used to further demonstrate that TMC improves particle efficiency. Consider a one-dimensional gamma distribution:

$$X \sim \Gamma\left(2, \frac{1}{3}\right)$$

The MC and TMC methods are used to generate particles from the gamma distribution. Figure 3 shows the sampling results from the two sampling methods. For the MC method, the selection of particles is random. Increasing the particle number allows for a better description of the gamma distribution, and this is the only way to mitigate the indeterminacy. For the TMC method, the position of particles is determined when the confidence and sampling interval are provided. The task of describing the probability distribution is transferred to the weights corresponding to the particles. The estimation results for the expectation errors of the Monte Carlo method for the gamma distribution

are shown in Figure 4. Considering the indeterminacy of MC, it is run 10,000 times and the RMSE is used to reflect the size of the error:

$$RMSE_m = \sqrt{\sum_{j=1}^{monte}\left(\sum_{i=1}^{m}x_i/m - 6\right)^2 / monte} \qquad (20)$$

where *m* denotes particle number and *monte* denotes the Monte Carlo number. The RMSE decreases as the particle number increases. Figure 4 shows that the RMSE is about 0.21 when the particle number is 400. For TMC, the relationship between the magnitude of confidence and the mean error is shown in Figures 5 and 6. The absolute expectation error decreases rapidly as the confidence increases. Meanwhile, the particle number increases slowly as confidence increases over a fixed sampling interval. The absolute expectation error can be reduced to 0.015 using only 75 particles for the confidence of 0.999, whereas the sampling interval is 0.4.

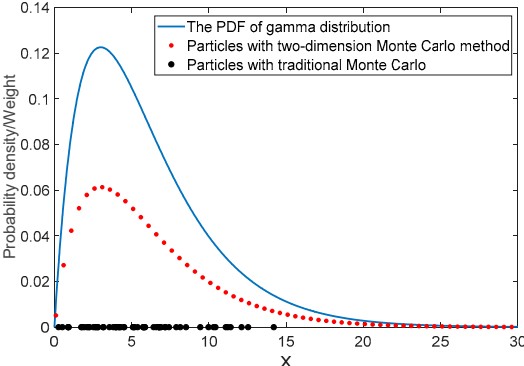

**Figure 3.** Sampling results from TMC and MC.

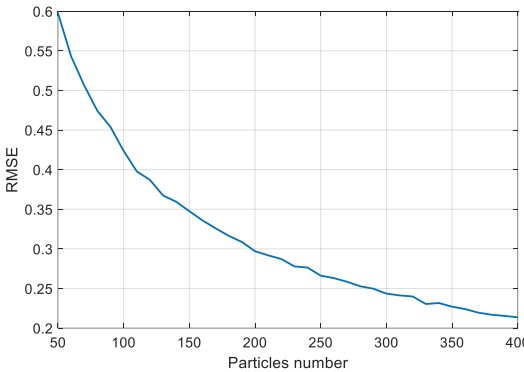

**Figure 4.** Variation of the expectation RMSE with the particle number for MC.

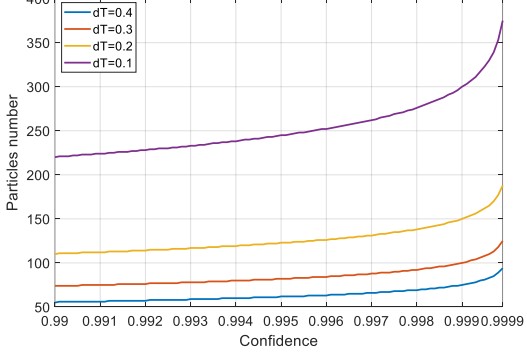

**Figure 5.** Variation of the particle number with confidence for TMC.

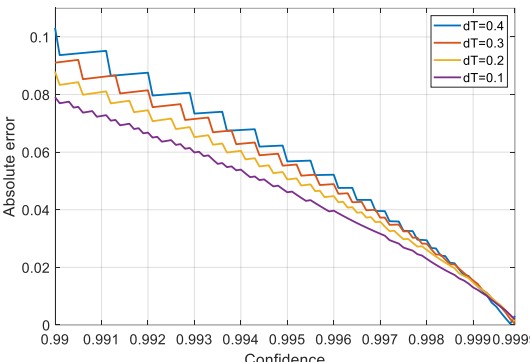

**Figure 6.** Variation of the absolute expectation error with confidence for TMC.

The results demonstrate that particles generated by TMC can describe the noise distribution more efficiently than particles generated by MC.

## 4. Proposed Filter Algorithm

Each of the efficient particles from TMC is a possible state estimation. The weight of each particle is the probability that the particle becomes the state estimation. The continuous probability distribution is discretized in terms of these particles and their weights. The TMCF is further designed as shown in Figure 7. The entire filter system can be divided into four parts: initialization, particle deployment, weight mixing and state estimation. In this section, the four parts are explained in detail and the TMCF algorithm is proposed.

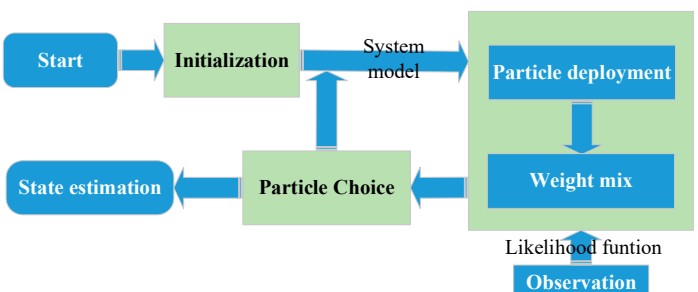

**Figure 7.** TMCF system block diagram.

*4.1. Initialization*

The target of initialization is to set several efficient particles to describe the initial probability distribution discretely to facilitate the subsequent filtering process. After the confidence $1 - \alpha$ and the sampling interval $dT_0$ are set, initial particles $X_0$ and their weights $w_0$ can be obtained in the confidence interval $c_0$ using the TMC method according to the known initial probability $p_0(x)$. Additionally, the confidence interval $c_\varepsilon$ for the system noise probability $p_u(x)$ of $1 - \alpha$ can be obtained according to Equation (8), and then the amplification of interval $\varepsilon$ is defined as:

$$\varepsilon = [column(c_\varepsilon)_1 - E(p_u(x)),\ column(c_\varepsilon)_2 - E(p_u(x))] \tag{21}$$

where $column(c_\varepsilon)_i$ denotes the $i$th column of matrix/vector $c_\varepsilon$ and $E(p_u(x))$ denotes the expectation of $p_u(x)$.

The real state $x_{real,0}$ now exists in the confidence interval $c_0$ for the probability $1 - \alpha$. $p_0(x)$ is described by $\{X_0,\ w_0\}$ for the accuracy of $dT_0$. The probability of each particle's existence is described by its weight.

### 4.2. Particle Deployment

The target of this step is to analyze the transition of the confidence interval from time step $t - 1$ to time step $t$, and then deploy particles. At time step $t - 1$, the confidence interval $c_{t-1}$ can be written as:

$$c_{t-1} = [\min(X_{t-1}), \max(X_{t-1})] \tag{22}$$

in terms of the principle of the TMC method. where $\min(X_{t-1})$ denotes the minimum value of each row of matrix/vector $X_{t-1}$ and $\max(X_{t-1})$ denotes the maximum value of each row of $X_{t-1}$. When the particles $X_{t-1}$ are transferred through the system model without system noise, the transferred particles can be expressed as:

$$\breve{X}_t = f(X_{t-1}) \tag{23}$$

Each particle in $\breve{X}_t$ then is considered to be a possible state estimation without system noise at time step $t$, and the probability of each particle $\breve{X}_t(i)$ is the weight of $X_{t-1}(i)$. Considering system noise, the confidence interval of each particle $\breve{X}_t(i)$ for confidence $1 - \alpha$ is

$$c_t(i) = \left[ \breve{X}_t(i) + column(\varepsilon)_1, \breve{X}_t(i) + column(\varepsilon)_2 \right] \tag{24}$$

where $c_t(i)$ denotes the confidence interval for confidence $1 - \alpha$ corresponding to $\breve{X}_t(i)$. Then, the complete confidence interval for $1 - \alpha$ can be obtained by calculating the union of all confidence intervals:

$$c_t = c_t(1) \cup c_t(1) \cdots \cup c_t(n_{t-1}) \tag{25}$$

where $n_{t-1}$ denotes the number of particles in set $X_{t-1}$.

For simplicity, the complete confidence interval also can be estimated roughly by:

$$\hat{c}_t = \bar{c}_t + \varepsilon \tag{26}$$

where $\bar{c}_t = \left[ \min\left( \breve{X}_t \right), \max(\breve{X}_t) \right]$.

As $\hat{c}_t \subset c_t$. Hence, confidence corresponding to the confidence interval $\hat{c}_t$ is greater than or equal to $1 - \alpha$. However, the amplification of the confidence interval might increase the number of deployed particles. Sometimes this phenomenon, particularly in the case of high dimensions, leads to too many particles, which might result in the failure of filtering.

Then, particles $X_t$ can be deployed according to the confidence interval $c_t$ or $\hat{c}_t$ and $dT_t$ at time step $t$. Generally, $dT_t$ is set to a constant vector:

$$dT_t = dT_0 \quad (t = 1, 2, 3 \cdots) \tag{27}$$

When the confidence interval is unstable (increases or decreases over time), a specific strategy corresponding to the specific system needs to be designed to change the size of the sampling interval.

In this step, the deployed particles $\bar{X}_t$ are distributed in this confidence interval uniformly, which is preparation for the subsequent step.

### 4.3. Weight Mix

In the weight mix step, the relationship between $X_{t-1}$ and $\overline{X}_t$ is analyzed to solve the prior weight corresponding to $\overline{X}_t$, the likelihood weight is calculated and the posterior weight is obtained.

As the distribution of the system noise is continuous, each particle in set $X_{t-1}$ might arrive at any particle in set $\overline{X}_t$, in theory. The probability of each particle in set $X_{t-1}$ arriving at each particle in set $\overline{X}_t$ can be expressed as:

$$\widehat{w}_t(i,j) = \frac{p_u\left(\overline{X}_t(j) - \breve{X}_t(i)\right)}{\sum\limits_{j=1}^{m_t} p\left(\overline{X}_t(j) - \breve{X}_t(i)\right)} \tag{28}$$

$$(i = 1, 2, \cdots n_{t-1}; \ j = 1, 2, \cdots m_t)$$

where $\widehat{w}_t(i,j)$ denotes the probability of the $i$th particle in $X_{t-1}$ being transferred to the $j$th particle in $\overline{X}_t$. $m_t$ denotes the number of particles in set $\overline{X}_t$. As shown in Figure 8, the prior weight is calculated as:

$$\widetilde{w}_t(j) = \sum_{i=1}^{n_t} w_{t-1}(i) \times \widehat{w}_t(i,j) \tag{29}$$

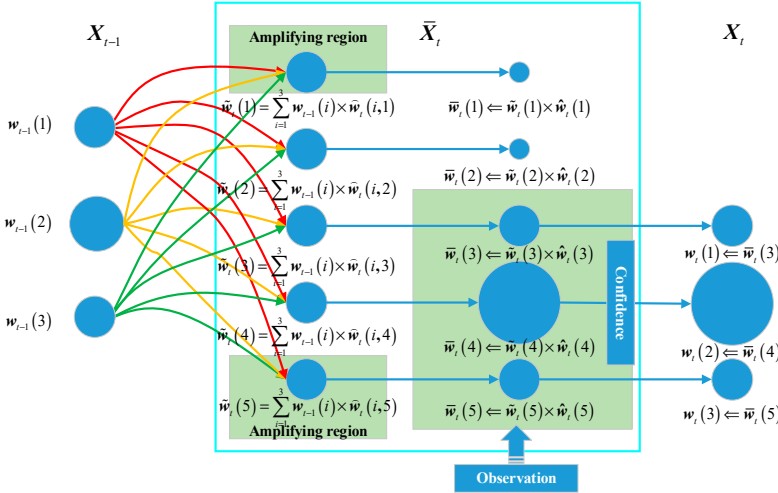

**Figure 8.** Schematic diagram of particle and weight changes.

The likelihood weight is written as:

$$\hat{w}_t(j) = p_v\left(h\left(\overline{X}_t(j)\right) - z_t\right) \tag{30}$$

Additionally, the posterior weight is mixed by:

$$\overline{w}_t(j) = \frac{\widetilde{w}_t(j)\hat{w}_t(j)}{\sum\limits_{i=1}^{n_t} \widetilde{w}_t(j)\hat{w}_t(j)} \tag{31}$$

Then, the posterior distribution of the state at time step $t$ is described by $\left\{\overline{X}_t, \overline{w}_t\right\}$ discretely.

### 4.4. Particle Choice and State Estimation

$\left\{ \bar{X}_t, \bar{w}_t \right\}$ describes the posterior distribution after the fusion of the prior distribution and the likelihood distribution. All the distribution information is concentrated in the weight $\bar{w}_t$, and the role of the particles $\bar{X}_t$ is to only occupy the distribution space. Generally, many very low weighted particles emerge after fusion. Additionally, these particles with very low weights have very little effect on the accurate description of the distribution, so they can simply be omitted. $n_t$ particles are chosen in the order of largest to smallest so that the sum of the weights of the $n_t$ particles is $1 - \alpha$:

$$\left\{ \bar{X}_t, \bar{w}_t \right\}_{n_t} \overset{1-\alpha}{\Longleftarrow} \left\{ \bar{X}_t, \bar{w}_t \right\}_{m_t} \tag{32}$$

Then, the weight is normalized:

$$w_t(i) = \frac{\bar{w}_t(i)}{\sum\limits_{i=1}^{n_t} \bar{w}_t(i)} \tag{33}$$

The state estimation can be obtained by

$$x_t = \sum\limits_{i=1}^{n_t} X_t(i) w_t(i) \tag{34}$$

In conclusion, the TMCF algorithm is summarized in Algorithm 1.

---

**Algorithm 1**

| | |
|---|---|
| 1 | Initialization: |
| 2 | Setting $1 - \alpha$ and $dT_0$ |
| 3 | Generate $\{X_0, \ w_0\}$ and $\varepsilon$ according to TMC method and Equation (21) |
| 4 | //Over all time steps: |
| 5 | **for** $t \leftarrow 1$ to $T$ **do** |
| 6 | Setting $dT_t = dT_0$, or other strategy is used to select $dT_t$ |
| 7 | Confidence interval choice according to Equation (25) or (26) |
| 8 | Particle deployment according to $dT_t$ |
| 9 | Weight fusion according to Equations (28)–(31) |
| 10 | Particles and their weights choice according to Equations (32) and (33) |
| 11 | State estimation according to Equation (34) |
| 12 | **End** |

---

## 5. Numerical Simulation

In this section, a two-dimensional linear system is used to assess the performance of the TMCF [43]:

$$\begin{bmatrix} x_1(t) \\ x_2(t) \end{bmatrix} = \begin{bmatrix} \cos(\theta) & -\sin(\theta) \\ \sin(\theta) & \cos(\theta) \end{bmatrix} \begin{bmatrix} x_1(t-1) \\ x_2(t-1) \end{bmatrix} + \begin{bmatrix} q_1(t-1) \\ q_2(t-1) \end{bmatrix} \tag{35}$$

$$z(t) = \begin{bmatrix} 1 & 1 \end{bmatrix} \begin{bmatrix} x_1(t) \\ x_2(t) \end{bmatrix} + r(t) \tag{36}$$

where $x_i(t)$ and $z(t)$ denote the state and observation at time step $t$, respectively; $q_i(t-1)$ denotes the system noise sequence at time step $t - 1$; and $r(t)$ denotes the observation noise sequence at time step $t$. In the two experiments, $\theta = \pi/18$ and the initial state is

$[1,1]^T$. The initial probability satisfies $p_0(x) \sim N(0, 0.1)$. It is well known that the KF is the optimal filter for a linear Gaussian system based on the Bayesian filter principle. Hence, the performance of the TMCF is first assessed in a linear and Gaussian system. The estimation results of the KF are used as a reference to evaluate the approximation degree of the TMCF algorithm and Bayesian filtering in the linear and Gaussian system. Because the TMCF algorithm is a filter based on the Monte Carlo principle, the performances of the TMCF algorithm and PF algorithm are compared in this experiment. Second, a heavy-tailed distribution (non-Gaussian environment) is considered in this linear system. The performance of the TMCF is compared with that of the KF and PF in this linear and non-Gaussian system. Four sets of parameters are selected for the TMCF algorithm, which are shown in Table 1. Two forms of mean square errors (MSE) are used to evaluate the performance of the algorithms:

$$MSE_1 = \sum_{i=1}^{T}(\hat{x}_i - x_{i,real})^2 / T \tag{37}$$

$$MSE_2 = \sum_{i=1}^{T}(\hat{x}_i - x_{i,KF})^2 / T \tag{38}$$

**Table 1.** Parameters of the TMCF.

| Parameter | $dT$ | $1-\alpha$ |
|:---:|:---:|:---:|
| 1 | 1.2 | 0.999 |
| 2 | 0.8 | 0.999 |
| 3 | 1.2 | 0.9999 |
| 4 | 0.8 | 0.9999 |

In this section, MATLAB is used to build the simulation environment. The performance (including the filtering precision, the number of samples, filter time) of the TMCF, KF and PF are verified and compared by this simulation environment. All the data are generated by the simulated program. The configurations of the simulation computer can be seen in Table 2.

**Table 2.** Configuration environment.

| CPU | Basic Frequency (GHz) | RAM (GB) | Windows Version | MATLAB Version |
|:---:|:---:|:---:|:---:|:---:|
| Intel(R) Core i5 | 1.70 | 16.0 | Windows 10 | R2018a |

### 5.1. Gaussian Distribution System

In this experiment, the Gaussian model is selected for both system noise and observed noise: $q_1(t) \sim N(0,1)$, $q_2(t) \sim N(0,1)$ and $r(t) \sim N(0,1)$.

PFs with 3000 and 5000 particles are used as the comparison algorithms of the TMCF. Figures 9 and 10 show that the deviation between each filtering result of the different algorithms and the KF results for $x_1$ and $x_2$, respectively. The results show that it is difficult for the PF to approximate the performance of the KF in the linear and Gaussian system. Compared with the KF, although the number of particles is 3000, the results of the PF deviate by about 0.15 from that of the KF in each filtering process. Meanwhile, as the number of particles increases dramatically, this deviation declines very slowly. The deviation is about 0.1 when the number of particles is 5000. This is caused by the indeterminacy of the Monte Carlo method. The indeterminacy is greatly reduced when the TMC method is used to generate particles. Using the TMC method, the results of the TMCF are very close to those of the KF. The difference between the TMCF and KF is less than 0.01 for all four parameters selected. The deviation decreases as the confidence increases

and the sampling interval decreases. Particularly, the deviation is less than 0.001 when the confidence is 0.9999 and the sampling interval is 0.8. Figure 11 shows that only about 40 particles need to be transferred in each filtering process for parameter 1, and the number of set particles is about 250. The number of particles required increases as the sampling interval decreases and the confidence increases. For parameter 4, the number of transferred particles is about 130 and the number of set particles is only about 800. Figure 12 shows the time consumed in each filtering process by the different algorithms on a computer using the same configuration. The computation time of the TMCF is much less than that of PF. Table 3 shows the filtering results of 5000 time steps processed by Equations (37) and (38). The $MSE_2$ is about 0.01 for PF with 5000 particles, and the computation time is about 0.1 s for each filtering process. The $MSE_2$ reaches $10^{-6}$ for the TMCF with parameter 4, and the computation time is only about 0.0035 s. The results demonstrate that the TMCF can approximate the KF algorithm better with fewer particles and less computation in linear and Gaussian systems compared with PF. Different from the KF, the TMCF does not use the propagation characteristics of the conditional means and covariances of Gaussian noise in linear systems. Therefore, this method is also applicable to non-Gaussian noise.

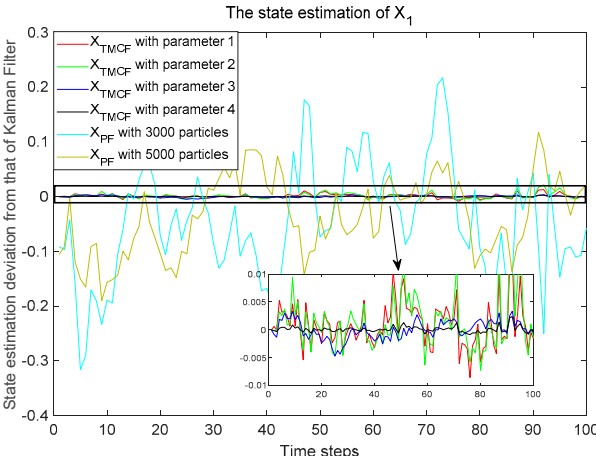

**Figure 9.** Difference between the state estimation results of the different algorithms and those of KF for $x_1$.

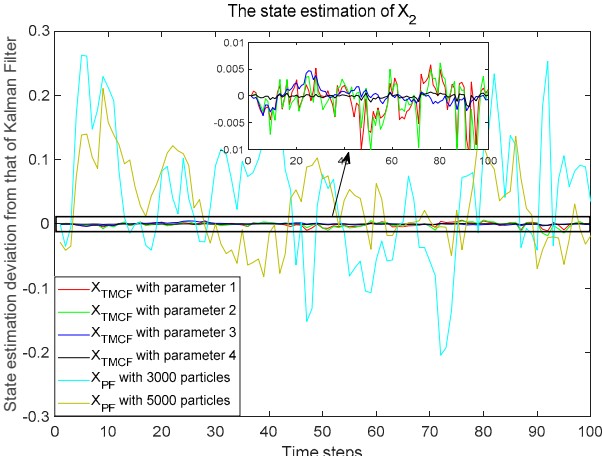

**Figure 10.** Difference between the state estimation results of different algorithms and those of KF for $x_2$.

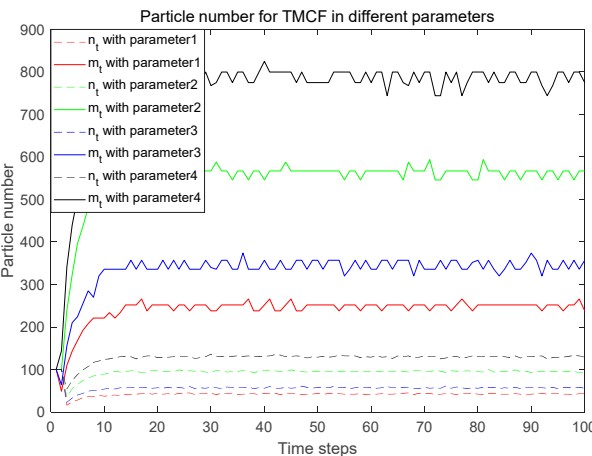

**Figure 11.** Number of particles required for the TMCF algorithm with different parameters.

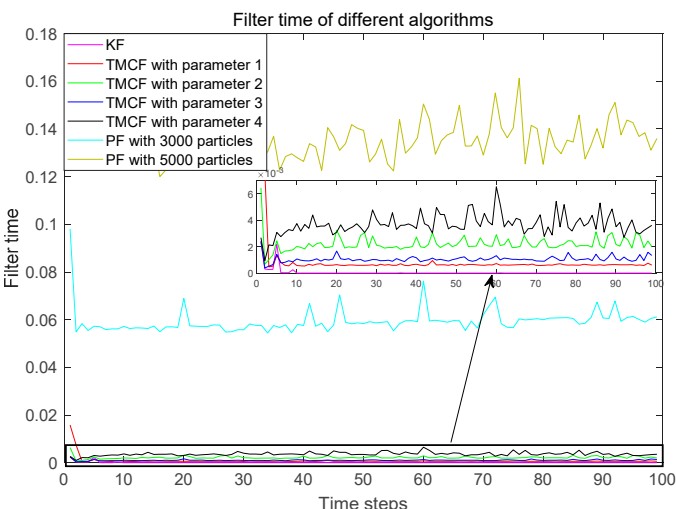

**Figure 12.** Computation time for the TMCF algorithm with different parameters.

**Table 3.** Performance of the different filter algorithms with different parameters in the linear/Gaussian system.

| Algorithm | Parameters | | | | $x_1$ | | $x_2$ | | Computation Time (s) |
|---|---|---|---|---|---|---|---|---|---|
| | $dT$ | $1-\alpha$ | $\overline{n_t}/\overline{m_t}$ | | MSE1 | MSE2 | MSE1 | MSE2 | |
| KF | - | - | - | | 3.421829 | 0 | 2.798550 | 0 | $3.069 \times 10^{-5}$ |
| PF | - | - | 3000 | | 3.456089 | 0.027105 | 2.822853 | 0.021010 | 0.0608210 |
| | - | - | 5000 | | 3.430909 | 0.015142 | 2.807029 | 0.012138 | 0.1387177 |
| TMCF | 1.2 | 0.999 | 42 | 248 | 3.423760 | $4.306 \times 10^{-4}$ | 2.799829 | $3.155 \times 10^{-4}$ | $6.282 \times 10^{-5}$ |
| | 0.8 | 0.999 | 95 | 560 | 3.422683 | $4.238 \times 10^{-4}$ | 2.798926 | $3.081 \times 10^{-4}$ | $2.101 \times 10^{-3}$ |
| | 1.2 | 0.9999 | 57 | 343 | 3.421978 | $9.444 \times 10^{-6}$ | 2.798610 | $7.098 \times 10^{-6}$ | $9.293 \times 10^{-4}$ |
| | 0.8 | 0.9999 | 130 | 782 | 3.421878 | $8.874 \times 10^{-6}$ | 2.798539 | $6.484 \times 10^{-6}$ | $3.511 \times 10^{-3}$ |

### 5.2. Gaussian Mixture Distribution System

In this experiment, the Gaussian mixture model is selected for both system noise and observed noise: $q_1(t) \sim 0.6N(0,1) + 0.4N(0,4)$, $q_2(t) \sim 0.6N(0,1) + 0.4N(0,4)$ and $r(t) \sim 0.6N(0,1) + 0.4N(0,4)$.

The noise model is shown in Figure 13. Similar to Table 3 for the previous experiment, Table 4 shows the filtering results of 5000-time steps processed by Equation (37). The $MSE_1$ of the PF with 3000 particles is greater than that of the KF. The performance of the TMCF with parameter 1 is better than that of PF with 5000 particles and KF. Meanwhile, the number of transferred particles is only 110 and the number of set particles is only 650. The computation time is 0.006 s, that is, much less than that of the PF. With the decrease of the sampling interval and the increase of confidence, the accuracy of filter is improved, and the computation time is increased. For the parameters 4 of TMCF, the accuracy of the TMCF is improved by 0.01, and the computation time reduced to 0.067 s from 0.20 s, comparing with the particle filter (5000 particles).

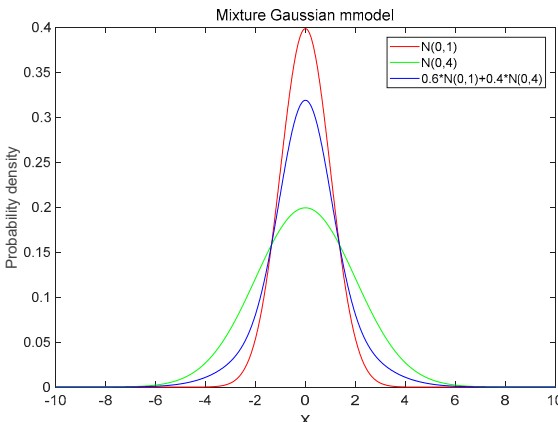

**Figure 13.** Probability model of Gaussian mixture noise.

**Table 4.** Performance of the different filter algorithms with different parameters in the linear/mixture Gaussian system.

| | Parameters | | | | MSE1 | | Computation Time (s) |
|---|---|---|---|---|---|---|---|
| | $dT$ | $1-\alpha$ | $\overline{n_t}/\overline{m_t}$ | | $x_1$ | $x_2$ | |
| KF | - | - | - | | 7.619301525 | 6.3982152 | $2.224 \times 10^{-5}$ |
| PF | - | - | 3000 | | 7.718597333 | 6.4887486 | 0.098728173 |
| | - | - | 5000 | | 7.589158022 | 6.3695680 | 0.204914013 |
| TMCF | 1.2 | 0.999 | 110 | 650 | 7.581596248 | 6.3654145 | 0.006746154 |
| | 0.8 | 0.999 | 250 | 1479 | 7.582551190 | 6.3659966 | 0.028529454 |
| | 1.2 | 0.9999 | 157 | 961 | 7.577101831 | 6.3616057 | 0.012136754 |
| | 0.8 | 0.9999 | 358 | 2184 | 7.576998566 | 6.3615759 | 0.067335938 |

## 6. Conclusions

The TMCF algorithm was proposed to overcome the challenge of the non-Gaussian filtering in this paper. First, the TMC method was proposed to sample particles in the confidence interval according to the sampling interval. The performance of the TMC method has been simulated and the property of the TMC method has been proved. Second, the TMCF algorithm was proposed by introducing the TMC method into the PF algorithm. Different from the PF, the TMCF algorithm completes the transfer of the distribution using a series of calculations of weights and particles were used to occupy the state space in the confidence interval. Third, Numerical simulations demonstrated that the MSE of the TMCF was about $10^{-6}$ compared with that of the Kalman filter (KF) in a two-dimensional linear/Gaussian system. In a two-dimensional linear/non-Gaussian system, the MSE of the TMCF for parameter 4 was 0.04 and 0.01 less than that of the KF and PF with 5000 particles, respectively. The single filter times of the TMCF and PF with 5000 particles were 0.006 s and 0.2 s, respectively.

In this paper, we have designed an improved PF algorithm, we called TMCF algorithm. In the non- Gaussian filter environment, it can not only improve the accuracy of filter, but also reduce the computation time. In the future development of new disciplines such as artificial intelligence, multi-sensor data fusion, and multi-target tracking, there are more and more types of data and more and more complex sources of data. the quality of nonlinear non-Gaussian filtering method becomes more and more important in data fusion. Our work lays a theoretical foundation for nonlinear/non-Gaussian filter and can be used to improve the filtering precision under the condition of reducing computation time in some non-Gaussian filter environments. In the future, we will try to apply the algorithm to the integrated navigation system to improve the positioning accuracy of satellite navigation.

**Author Contributions:** Conceptualization, R.X. and X.Q.; methodology, R.X.; software, X.Q.; validation, R.X., X.Q. and Y.Z.; formal analysis, X.Q.; writing—original draft preparation, X.Q.; writing—review and editing, R.X. and X.Q.; supervision, Y.Z.; funding acquisition, R.X., X.Q. and Y.Z. All authors have read and agreed to the published version of the manuscript.

**Funding:** This work was supported in part by the National Key Research and Development Program of China, grant number 2017YFB0503400, and in part by the National Natural Science Foundation of China, grant numbers U2033215, U1833125 and 61803037.

**Conflicts of Interest:** The authors declare no conflict of interest.

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
