# Peer review of "Two-Dimensional Monte Carlo Filter for a Non-Gaussian Environment"

_electronics, doi:10.3390/electronics10121385_

Round 1

Reviewer 1 Report

Dear Authors.

This paper (Two-dimensional Monte Carlo Filter for a Non-Gaussian Environment) provides little information. But, it is good motivation and result. So, I give Major Revision.

The strength of the paper included: the topic is good and interesting.

#1. Abstract
Look at MDPI guideline
https://www.mdpi.com/journal/electronics/instructions
-Abstract: The abstract should be a total of about 200 words maximum. The abstract should be a single paragraph and should follow the style of structured abstracts, but without headings: 1) Background 2) Methods 3) Results and 4) Conclusion.

#2.
What kind of Simulation/Coding language did you use?
-Please add/more version and description.
-Simulation/Coding language environment.

#3. Introduction
Check at MDPI guideline
https://www.mdpi.com/journal/electronics/instructions

•Introduction: The introduction should briefly place the study in a broad context and highlight why it is important. It should define the purpose of the work and its significance, including specific hypotheses being tested. The current state of the research field should be reviewed carefully and key publications cited. Please highlight controversial and diverging hypotheses when necessary. Finally, briefly mention the main aim of the work and highlight the main conclusions. Keep the introduction comprehensible to scientists working outside the topic of the paper.

#4.
Does the introduction provide sufficient background and include all relevant references? Must be improved
Is the research design appropriate? Can be improved
Are the methods adequately described? Can be improved
Are the results clearly presented? Can be improved
Are the conclusions supported by the results? Must be improved

#5. Contribution
-It is almost impossible to understand the contribution of the paper.

#6. Completeness and Related Work are good.
-How about making a "Related Work" chapter? 
-Also, improve more related works, more to 10 new papers published from 2019~2021 by major publishers such as IEEE, ACM, Springer, Elsevier, MDPI, and Wiley. (Need new papers)

#7. English language:
Moderate English changes required.

#8. Results
-Provide a concise and precise description of the experimental results, their interpretation as well as the experimental conclusions that can be drawn.

#9. Conclusion Need:
-Future work" write more. (Must be improved)
-Conclusions supported by the results. (Write more)
-This section is not mandatory, but can be added to the manuscript if the discussion is unusually long or complex.

#10. Scientific Soundness: Low

Reviewer 2 Report

Dear Authors,

The subject of the article is very interesting, although I have two serious doubts:

  • is the subject matter presented in the manuscript consistent with the line of the Electronics journal?
  • the analysis contained is purely theoretical and does not go beyond the scope of computer simulations.

Regarding the first remark - it is up to the Editor to decide whether the subject matter is compatible with the content of the manuscript.

Regarding the second doubt - I believe that it is worth supplementing the manuscript with issues related to the practical aspects of implementing the TMC algorithm in ELECTRONIC devices and making a comparison of e.g. the necessary computing powers against KF and PF. Another issue that may be raised is the attempt to process data obtained from a real system (e.g. electronic sensors).

The summary is very short and general. It does not refer to quantitative results.

Editing Notes:

  • references in the text to tables are in the form of  e.g. "Table IV" - it is inconsistent with the MDPI format
  • on lines 135-150 there is a nomenclature. It should be placed at the beginning or the end of the article as an element separated from the article structure. Relatively, individual descriptions of quantities may already be placed in the text, directly the first time they are used.

Reviewer 3 Report

This work provides an answer to the problem of non-Gaussian filtering.

In this paper, the two-dimensional Monte Carlo  methodis is proposed to improve the efficacy of the particle filter.

This work is interesting but there is a lack of justification for the need to use such an approach.

It is important that the authors situate themselves in relation to first [1, 2] and second generation [3] wavelet-based filtering which requires no assumptions about the nature of the noise, and that they cite these 3 key references.

[1] Ouahabi, A. Signal and Image Multiresolution Analysis, 1st ed.; ISTE-Wiley: London, UK, 2012.

[2] Ouahabi, A. A review of wavelet denoising in medical imaging. In Proceedings of the International Workshop on Systems, Signal Processing and Their Applications (IEEE/WOSSPA’13), Algiers, Algeria, 12–15 May 2013; pp. 19–26.

[3] Sidahmed, S. et al. Nonparametric denoising methods based on contourlet transform with sharp frequency localization: Application to electron microscopy images with low exposure time. Entropy 201517, 2781–2799.

Moreover, the authors use some unconventional notations such as the set of reals, the double bar on a variable... It would be more appropriate to go back to classical notations for a better readability of the article.

The references are not always presented according to the standard imposed by electronics MDPI.

Round 2

Reviewer 1 Report

Dear Authors. 
The revision adequately address the concerns expressed in last review. So, I recommend that this revised manuscript can now be recommended for publication (Accept as is).

Reviewer 2 Report

Dear Authors,

I accept you responses to my comments 1-2. Thanks for making corrections on comments 3-5. I accept the manuscript as it is presented now.

Reviewer 3 Report

The authors corrected their manuscript, which improved its content.